# Embryonic Kidney Development, Stem Cells and the Origin of Wilms Tumor

**DOI:** 10.3390/genes12020318

**Published:** 2021-02-23

**Authors:** Hao Li, Peter Hohenstein, Satu Kuure

**Affiliations:** 1Stem Cells and Metabolism Research Program, Faculty of Medicine, University of Helsinki, FIN-00014 Helsinki, Finland; hao.li@helsinki.fi; 2Department of Human Genetics, Leiden University Medical Center, 2300 RC Leiden, The Netherlands; p.hohenstein@lumc.nl; 3GM-Unit, Laboratory Animal Center, Helsinki Institute of Life Science, University of Helsinki, FIN-00014 Helsinki, Finland

**Keywords:** kidney, organogenesis, pediatric cancer, stem cells, differentiation, self-renewal

## Abstract

The adult mammalian kidney is a poorly regenerating organ that lacks the stem cells that could replenish functional homeostasis similarly to, e.g., skin or the hematopoietic system. Unlike a mature kidney, the embryonic kidney hosts at least three types of lineage-specific stem cells that give rise to (a) a ureter and collecting duct system, (b) nephrons, and (c) mesangial cells together with connective tissue of the stroma. Extensive interest has been raised towards these embryonic progenitor cells, which are normally lost before birth in humans but remain part of the undifferentiated nephrogenic rests in the pediatric renal cancer Wilms tumor. Here, we discuss the current understanding of kidney-specific embryonic progenitor regulation in the innate environment of the developing kidney and the types of disruptions in their balanced regulation that lead to the formation of Wilms tumor.

## 1. Introduction

Stem cells form a fundamental basis not only for normal development and tissue homeostasis but also for tumorigenesis. The adult mammalian kidney is mostly devoid of stem cells and is thus considered a non-regenerative organ, especially when regeneration is defined by the ability to generate new nephrons and regain filtration capacity [1]. Contrary to this, the embryonic kidney hosts at least three types of lineage-specific stem cells (referred to here as progenitors) that give rise to the ureter and collecting duct system, nephrons, and the connective tissue of the stroma [2,3]. These progenitor cells have raised substantial interest for use in regeneration-based renal medicine, but the safety of the undifferentiated cell maintenance in postnatal kidneys is a significantly less discussed topic. 

An integral part of the safety assessment is a thorough understanding of tissue-residing progenitor regulation. The mechanisms guiding the maintenance and differentiation of tissue-specific progenitors are aberrantly reactivated in many cancers [4]. This is especially obvious in embryonic-derived tumors such as Wilms tumor (WT; Figure 1), medulloblastoma, and retinoblastoma. 

Nephrons—the functional filtration units of mammalian kidneys—and renal stroma are derived from metanephric mesenchyme containing distinct progenitor pools for both lineages [5,6]. Metanephric tissue is normally lost before birth in humans but remains part of the undifferentiated nephrogenic rests in Wilms tumor patients [7,8]. Understanding the differences between normal tissue-residing progenitors and cancer-causing stem cells is required to set the stage for better characterization of transformations that turn normal kidney progenitors into Wilms tumor stem cells. This may essentially help with predicting individualized disease prognosis and tumor chemosensitivity, thus facilitating better patient stratification and identification of the patients requiring aggressive treatment strategies [9]. This review gives an overview of kidney development followed by a summary of how collecting duct and nephron progenitors contribute to normal renal morphogenesis in order to better facilitate the understanding of their fundamental similarities and differences with Wilms tumor biology.

## 2. Wilms Tumor

Wilms tumors (Figure 1) are one of the most common solid tumors in children with an incidence of 1:10,000, usually appearing before the age of five. Although good treatment options exist for some histological tumor types and their survival rates are good, other types, like anaplastic Wilms tumors, still have a 5-year survival of only 50%. Therefore, there remains a clear clinical need for the improvement of Wilms tumor therapeutic options; to achieve this, a better basic understanding of these tumors is essential.

As reviewed extensively elsewhere [7], Wilms tumors have been intriguing clinicians, pathologists, and cancer geneticists for a long time. The tumors are the direct result of problems during the embryonic development of the kidney, and it was one of the cancer types based on which Alfred Knudson developed his two-hit model for tumor suppressor genes [10]. It is widely believed that nephrogenic rests—embryonic tissues that are retained in the postnatal kidney—are the precursor lesions of Wilms tumors, and indeed, the tumors themselves contain cell types normally found only in the embryonic kidney. A subset of cases, the ones caused by the loss of the *Wilms Tumor1* (*WT1*) tumor suppressor gene, additionally present with the development of ectopic, non-renal tissues, especially striated muscle. All this points to the disruption of normal development as the primary cause of Wilms tumors. We therefore need to understand normal kidney development in order to really understand Wilms tumors and exploit this understanding to develop better and more specific therapies. At the same time, the primary defect leading to Wilms tumors could be considered a naturally occurring model to understand normal kidney development.

## 3. Embryonic Kidney

Kidney morphogenesis is a classic example of balanced reciprocal tissue interactions [11,12,13]. Much of our basic understanding of how the kidney develops derives from the classical in vitro tissue recombination/induction experiments in different model organisms that are complemented with in vivo gene inactivation studies in mice [14,15]. These experiments have demonstrated that the mammalian kidney derives from the intermediate mesoderm, which gives rise to the three spatiotemporally distinct kidneys called pro-, meso-, and metanephros [15,16,17]. Organogenesis of the metanephros, the definitive kidney, utilizes epithelial ureteric bud (UB) branching morphogenesis for growth and patterning of the future organ, while nephron differentiation occurs in the nascent metanephric mesenchyme that surrounds each UB tip (Figure 2). Each newly formed UB is responsible for keeping the majority of the metanephric mesenchyme population intact while inducing its subpopulation to undergo stepwise mesenchyme-to-epithelium transformation in the armpits of T-bud epithelium to form functional nephron [18]. Orchestrated repetition of this cycle in a highly regulated manner ensures the maintenance of all relevant cell types until the completion of kidney organogenesis. Renal stroma is a part of the mesenchymal population that caps the nephron-forming mesenchyme and is critical not only for the formation of mesangial cells and interstitium, but also actively participates in the regulation of branching morphogenesis, and the proper differentiation of nephrons and vasculature [19,20,21,22,23,24]. While the innervation and vascular network formation are essential features of functional kidney development and recent studies indicate a presence of endothelial precursors in embryonic kidney, these topics are not discussed here (for insights, see [25,26,27,28,29,30,31,32,33]).

### 3.1. Kidney Induction

Renal development begins by the mesenchyme-to-epithelium transition through which the intermediate mesoderm differentiates into the epithelial nephric (Wolffian) duct that subsequently grows towards the posterior end of the embryo and simultaneously specifies the metanephric mesenchyme in the posterior part of the embryo [34]. After connection to the cloaca (the future urogenital sinus), which occurs at embryonic day 10.5 (E10.5) in mice, the induction of the definitive kidney takes place when the epithelial nephric duct forms a single bud that grows into adjacent metanephric mesenchyme to establish the ureteric bud (UB) [35,36]. Kidney development in humans begins around gestational days 28 to 30. Over the last couple of years, great leaps have been made in understanding the detailed developmental timing of human kidney differentiation and its molecular and morphological mechanisms [37,38,39]. These studies support the previous view established based on earlier studies that despite some differences, renal differentiation in mice and humans is well-conserved.

It is generally accepted that after kidney induction, the budding and the first UB branching event are cellularly and molecularly distinct from the subsequent branching events, which are closely interlinked with nephrogenesis [40,41]. For example, the nephric duct giving out the initial UB and the UB itself are composed of a pseudostratified epithelium, and the transcriptional profile of early metanephric mesenchyme diverges from that of the cap mesenchyme representing the nephron progenitor population (https://www.gudmap.org/chaise/record/#2/RNASeq:Replicate/RID=16-2PQ2 (accessed on 27 January 2021)) [42,43,44,45,46]. The initial UB formation is followed by elongation, subsequent enlargement of the UB tip into an ampulla, and finally bifurcation of the ampulla into T-shaped branches [41]. Somehow, through mostly unknown mechanisms, formation of the first T-bud establishes a molecular machinery that is able to maintain the nephrogenic program in the nascent metanephric mesenchyme until approximately gestational week 30 to 32 in humans and early postnatal days in mice, both of which are beyond the cessation of branching morphogenesis itself [47,48,49,50].

The key transcription factors required for kidney induction include Eya1, Hox11 paralogues A and D, Pax2, Sall1, Six1 and Wt1 [45,51,52,53,54,55], and their roles have been extensively reviewed elsewhere [56,57]. It is also well-established that the simultaneous activation of receptor tyrosine kinase signaling, especially downstream of the glial cell line-derived neurotrophic factor (GDNF), rearranged during transfection (RET), and fibroblast growth factor (FGF) receptor, together with inhibitory actions of bone morphogenetic protein signaling are required for UB formation and metanephric mesenchyme induction [43,58,59,60]. Less is known about the detailed regulatory relationships between signaling pathways and transcription factors, but of the above-mentioned transcription factors, Eya1, the Hox11 paralogues, Pax2, and Six1 are required for Gdnf expression and thus the activation of RET signaling, which in turn mediates its effects through transcription factors Etv4 and -5 [43,61,62,63]. Substantially more work is needed to map the exact regulatory roles of signaling pathways, transcriptional targets and phosphoproteomic control of cellular events during the early kidney induction.

### 3.2. Ureteric Bud Branching Morphogenesis Orchestrates Embryonic Kidney Growth and Nephron Formation

After the first UB bifurcation and establishment of metanephric mesenchyme, branching continues for 12 successful cycles to complete 85% of branching events by E16.5, after which the UB trunk elongation phase and completion of the final branch generations occur before birth [49,50,64]. Cellularly, UB branching happens through proliferation in both the tips and trunks, with the difference that the cell cycles are significantly faster in the tips than in the trunks [63,65]. Cell divisions in the UB employ a unique process of luminal mitosis, where epithelial cells partially delaminate from the epithelial sheet to divide in the luminal site followed by re-insertion of the mother and daughter cells back to the epithelium a few cells apart from each other [66]. Such a process requires extensive cellular movements, which are made possible through the dynamic and constant remodeling of adhesions and the actin cytoskeleton required for normal branching to progress [42,63,67,68,69]. In addition to proliferation, UB trunk elongation and thinning occur through oriented cell divisions and cellular rearrangements known as convergent extension [70,71], which also likely continues to contribute to kidney growth after birth.

A combination of mathematical modeling and highly accurate imaging has generated new information that challenges the traditional view of the UB branching pattern being a stereotypic repetition of dichotomous bifurcations with occasional tip trifurcation and very rare lateral branching events mainly observed in cultured kidneys [50,72]. A bi-phasic, time-dependent branching pattern was recently suggested based on the findings that rapid and reproducible branching during the early kidney development (up to E15.5 in mice) is followed by more non-stereotypic branching events closer to birth when the rate of new tip formations is also significantly decreased [64]. This is suggested to progressively generate variability in the 3D structure of the UB tree. Support for asymmetry in UB branching exists [73], but the signals that contribute to the slowing down and eventual cessation of UB morphogenesis remain elusive.

## 4. Renal Progenitor Populations

The embryonic kidney, unlike in its mature form, contains progenitor cells that can assemble the entire collecting duct, the connective tissue of stroma, and the whole nephron with its functional segments (Figure 2) [43,74,75]. The most UB tip-adjacent metanephric mesenchyme, termed ‘cap mesenchyme’, is the source of nephron progenitors—a cell population that self-renews during each round of UB tip bifurcation and is maintained until late gestation in human and early postnatal stages in mice, after which this source of new nephrons is irreversibly lost [13,47,50]. Nephron progenitors, which surround each UB tip, are the best characterized renal progenitor population and are discussed separately in a designated section. UB tips host another embryonic progenitor population, which is capable of populating the entire collecting duct system of the mature kidney but is also already permanently lost in utero. Self-renewing stromal progenitors surround the nephron progenitor population and differentiate into mesangial cells and renal stromal lineages (Figure 3) [75].

## 5. Collecting Duct Progenitors

It has been known for a long time that GDNF expressed by the nephron progenitors in the metanephric mesenchyme is both required and sufficient for UB outgrowth from the nephric duct [13,76,77,78,79,80]. More recently, GDNF-activated RET signaling has been demonstrated to coordinate cellular events related to collecting duct progenitor behaviors [43]. The identification of Etv4 and Etv5 as transcription factors that mediate the GDNF/RET signaling effects to the target cells in the UB tips together with genetic labelling experiments demonstrated that a substantial number of cellular movements constantly occur not only within the UB tips but also from the tips to the trunk regions [41,42,61,62,63,81]. It is clear from the chimera experiments that the wild type-derived epithelial cells are competent to populate UB tips and trunks, while RET signaling-deficient cells failed to settle in the tips. Thus, GDNF/RET signaling influences the cellular movements and the position of an individual cell in a given UB, suggesting not only that the way cells move within the epithelium influences UB branching morphogenesis but also that the cell’s localization within the UB greatly influences its potency.

### GDNF/RET Signaling Regulates Collecting Duct Progenitors through MAPK/ERK Activity

Further insight into the function of GNDF in UB cell fate regulation came from the studies utilizing a Gdnf mouse model, which lacks the gene’s 3’ untranslated region (3’-UTR) [82]. Insertion of a strong bovine growth hormone polyA signal after the stop-codon in Gdnf endogenous locus abolishes normal 3’UTR function and results in the excess production of endogenous GDNF at the mRNA and protein levels (Gdnf-hypermorphic allele). The increased expression is likely due to the lack of binding sites for microRNA and other RNA binding proteins that are normally present in such regulatory regions [82,83]. The elevated but spatially intact expression of GDNF in the mutant mice results in severely expanded UB tips with short trunks that gave rise to short and expanded ureters with misplaced connections to the bladder [84]. The analysis of mutant kidneys showed that the formation of an enlarged primary UB is at least partially attributed to the transiently increased mitosis in the caudal nephric duct at E10.5, the stage when primary UB formation begins. Thereafter, the mitotic index of UB tip epithelial cells rapidly normalizes simultaneously with a surge of apoptosis in the UB lumen. This may suggest that the kidney has an inherent mechanism that tries to rehabilitate the normal morphology under pathological conditions. Cellular tracking experiments revealed the hindrance of emigration in the tip epithelial cells, which got stuck in the tips and failed to populate and elongate the UB trunks. This demonstrates that GDNF supports collecting duct progenitor expansion as UB tips remain enlarged due to the emigration defect throughout kidney morphogenesis.

Our own results show that GDNF affects collecting duct progenitor cells via activation of mitogen-activated protein kinase (MAPK)/extracellular signal-regulated kinase (ERK) [84]. This is supported by the earlier RET mutant models, where MAPK/ERK activation was blocked [85,86]. Only MAPK/ERK inhibition, not PI3K/AKT or SRC inhibition, rescues the UB tip morphology and trunk length in kidneys with endogenously increased GDNF. Live-imaging of developing kidneys expressing a fluorescence resonance energy transfer (FRET)-based biosensor for ERK has demonstrated the dynamic activation pattern with significant heterogeneity not only between tissues (UB tips vs. cap mesenchyme) but also among seemingly homogenous cell populations [87]. Heterogeneity in MAPK/ERK activation is remarkably obvious in the UB tips, where high and low MAPK/ERK activation is seemingly randomly scattered among the epithelium, suggesting a role for cell sorting. UB-specific genetic MAPK/ERK inactivation (Hoxb7Cre;Mek1^fl/fl^;Mek2^-/-^) shows a completely opposite phenotype to expanded UB tips in Gdnf-hypermorphic kidneys, as MAPK/ERK-deficient tips remain thin and fail to expand into ampullae structures [88]. This demonstrated the essential requirement of MAPK/ERK activation for new branch formation in UB tips, which elongate but very rarely change growth direction, resulting in an oversimplified UB tree and renal hypodysplasia. Molecularly, MAPK/ERK activity appears important not only for the G-to-S cell cycle phase progression but also normal PAXILLIN- and E-CADHERIN-mediated cellular adhesions. Given the importance of MAPK/ERK and E-CADHERIN in embryonic stem cells [89,90,91,92], future experiments addressing their regulatory relationships in the context of kidney development may provide interesting insights into collecting duct progenitor regulation. 

Based on the observation that UB branching takes place only in the tips in most cases and always gives rise to new tips with a similar potential to branch, the studies described here first established a hypothesis that UB tips are different from the trunks. The elaborated analysis of the imaging studies then confirmed that UB tips are the sites where the progenitor population for collecting ducts and ureter reside. Further genetic studies have revealed that Notch signaling is required for patterning the distribution of FOXI1+ principal and AQP2+ intercalated cells within the mature collecting duct while several additional genes including at least methyltransferase Dot1l, and transcription factors p63 and Tfcp2L1 contribute to the balanced differentiation of each cell type [93,94,95,96,97,98,99,100] (for a more detailed overview, see, e.g., [101,102]). It remains to be studied whether collecting duct progenitor maintenance and loss before birth has any link to the previously reported bi-phasic and time-dependent UB branching topology [64]. 

## 6. Stromal Progenitor Cells

The renal stroma is comprised of the interstitium, mesangium, and pericytes, derived from the FOXD1-positive self-renewing progenitor population [103,104]. Stromal progenitors also differentiate into mural cells of the kidney arteries and arterioles, as well as the mesangial cells of the glomerulus, thus contributing significantly to the nephron functions. In addition to essential transcriptional regulation provided by at least FoxD1, FoxG1, Gata3, and Pax2, stromal progenitors are dependent on Notch signaling [6,19,105,106,107]. In particular, Pax2 appears to form a boundary between nephron and stromal progenitors to critically repress stromal identity, while GATA2 and RBP-J/Notch signaling, independent of each other, are needed for proper renal vasculature development. More recent single-cell transcriptomic analysis of the FOXD1 lineage from E18.5 mouse embryos showed that, by that stage, the lineage can be subdivided into 17 separate clusters of cells indicating a remarkable cellular heterogeneity with different transcriptional programs driving gene expression in these clusters [108]. Reanalysis of previously generated human embryonic kidney single-cell data confirmed that this is not a mouse-specific phenomenon, as even in human data, which is not specifically selected for the stromal lineage, 13 different stromal clusters could be identified. 

Many roles of the stromal lineage are found in the communication with the other lineages. Early data identified a retinoic acid-mediated signal from the stroma to RET in the ureteric bud that controls the branching of the ureteric bud [22]. A branching phenotype, linked to the downregulation of Aldh1a2 (an essential component of the retinoic acid synthesis pathway) and Ret was also observed in the stromal progenitor-specific knockout of Wt1, although the disturbed branching was only observed in later-stage embryonic kidneys [23], suggesting that this is not necessarily the role of the stromal progenitors but more of later cell types in the stromal lineage. On the other hand, ablation of the Foxd1-positive stromal progenitors results in a block of the nephron progenitor differentiation and instead results in an expanded cap mesenchyme through the loss of a FAT4-YAP/TAZ-mediated signal that fine-tunes the Wnt9b response in the nephron progenitors [109]. Other data suggests FAT4 might signal via DCHS1 rather than YAP/TAZ, as the conditional knockout of Dchs1 in the nephron progenitors resulted in a comparable enlargement of the cap mesenchyme, but interestingly also in the reduced branching of the ureteric bud [110,111]. These branching phenotypes are mediated through the direct interaction of FAT4 and DCHS1 with RET, with Fat4 loss resulting in an overactive RET-GFRA1-GDNF cascade [21]. Finally, the Foxd1-Cre-mediated loss of Sall1 in the stromal compartment results in expanded cap mesenchyme, potentially through direct control of Fat4 expression by SALL1 [112]; in this case, effects on the ureteric bud were not studied. 

It is clear that the cells from the stromal lineage have direct effects on the epithelial (ureteric bud) and nephrogenic lineages, thereby enabling a coordinated development of the different lineages to form a functional kidney. Whether these functions are executed by the stromal progenitors themselves or by later cell types in this lineage remains to be determined, but the detailed analysis of the heterogeneity of this lineage discussed above might enable the identification of new marker genes that can be used as Cre drivers to study these phenotypes in more detail. 

## 7. Nephron Progenitors

After the establishment of metanephric mesenchyme, it first functions to create a specific environment for the primary UB to form at exactly the correct position [113,114,115,116,117,118,119]. The metanephric mesenchyme then promotes initiation of UB branching morphogenesis, which is required for its own survival and organization into a nephrogenic niche providing cells for nephrogenesis until the cessation of the nephrogenic program [5,48,120]. The nephrogenic niche (Figure 1) is a cohesive structure of three-to-five cell layers, where the first cell layer intimately interacts with the UB epithelium, and the rest of the population is surrounded by other progenitors and stromal cells. Live imaging of the niche provides evidence that nephron progenitors are highly motile and actively interacting with all other progenitors and can even jump from one niche to another [37,121,122].

Due to the reiterative UB branching, the nephrogenic niche faces a continuous morphological challenge. First, the undifferentiated nephron progenitor population, which is initially a uniform niche surrounding the UB tip, needs to split in two distinct populations upon tip bifurcation. After this, the progenitor cells must remain in contact with the ever-elongating, newly generated tips. The connection established by the most UB-adjacent nephron progenitor subgroup to the UB tip cells is vital for the integrity of the entire niche. Molecularly, it is mediated at least through integrin α8 (ITGα8), which is upregulated upon contact and strongly localizes to the proximal membranes touching the tip epithelium where it interacts with NPNT (nephronectin) ligand present in the UB [123,124]. Nephron progenitors express numerous additional adhesion proteins, e.g., NCAM1, CDH2, -11 and CTNND1, which are all likely to contribute to niche cohesion [69,125,126].

The second challenge is active cell segregation within the nephrogenic niche, which takes place only to certain cells subjected to differentiation signals in the environment that is full of overlapping cues simultaneously promoting both self-renewal and differentiation. Third, the total niche numbers and their combined volumes increase towards the end of organogenesis, but the amount of nephron progenitor cells in each individual niche declines simultaneously. While this is all happening, each niche must sustain sufficient progenitors through ample proliferation, although cell cycle length increases over the time [48,50,127].

The individual progenitors exhibit columnar alignment and elongated cell shape, with Golgi apparatus located in the distal half of the cell, but upon detachment from the tip, the progenitors adopt a rounder shape likely reflecting the dynamic changes in their cell adhesions [122,126,128]. Towards the end of their existence, the nephron progenitor localization becomes restrained to positions more lateral to the tip [48], suggesting a decrease in the UB’s regulatory effect on niche organization. Moreover, clear signs of progressive aging are reported in the old vs young nephron progenitors including differences in ribosomal biogenesis, cell cycle length and extracellular matrix composition [127].

### 7.1. Molecular Determinants of Nephron Progenitors

Like collecting duct progenitors, nephron progenitors also represent a heterogeneous population. Progenitors display differences especially in their cell cycle lengths and expression profiles of, e.g., sine oculis-related homeobox 2 (SIX2) and Cbp/p300-interacting transactivator 1 (CITED1), which associate with the differentiation status of any given progenitor [5,129,130]. The exact mechanisms through which spatially distinct progenitor subgroups are formed needs further investigation, but it appears that nephron progenitors are not clonally expanded as each daughter cell is rather stochastically dispersed after cell divisions [50,122,131]. The most undifferentiated NPs express high levels of SIX2 and CITED1, and they self-renew slowly through a prolonged cell cycle. The committed progenitors no longer express CITED1 and have lower SIX2, cycle faster and are susceptible to nephron induction [50,132]. While Six2 is required for keeping nephron progenitors undifferentiated, Cited1, even in the absence of its close family member Cited2, appears non-essential for the progenitor behaviors [5,133,134]. 

Recent reports suggest a certain flexibility in nephron progenitor commitment as those progenitors expressing Wnt4, and thus molecularly induced for the differentiation path, can still escape from the nephron precursor structures to re-join the undifferentiated niche [135]. These escapees behave distinctively from the majority of induced progenitors as they downregulate Wnt4 expression and re-acquire a progenitor profile that is able to support their long-term self-renewal capacity [135,136]. This, together with high overall motility, supports the view of complex molecular networks in nephron progenitor regulation, where signaling levels may play bigger roles than previously appreciated. Indeed, changes in signaling levels have also been suggested to contribute to the cessation of nephrogenesis, during which the nephron progenitors exit the undifferentiated niche with accelerated speed without showing a dramatic decrease in proliferation or increase in apoptosis [13,47,48,49,50,127,137,138,139]. These data indicate that the progenitor pool becomes exhausted due to increased differentiation, which depletes the nephron progenitor niche by postnatal day four in mice and during the last gestational weeks in humans.

### 7.2. Nephron Progenitor Maintenance Depends on Classical Signaling Pathway Activities

Over the past twenty or so years, the transcriptional regulation of nephron progenitor maintenance and the inductive cues triggering nephron progenitor differentiation towards the nephron fate have been the focus of intensive research [34,56,140,141,142]. The majority of signaling pathways that are active during embryogenesis are also involved in nephron progenitor regulation [3,18,58,143]. The distinct roles of intracellular cascades activated downstream of ligand-receptor interactions are only about to be revealed [4,144]. For the sake of this review, we will focus on the roles of WNT/β-catenin, IGF2/FGF, microRNAs and mTOR-induced signaling in nephron progenitor maintenance and exhaustion.

### 7.3. The Wnt Pathway

Classical induction experiments and the use of chemical agonists/antagonists have demonstrated that transient WNT/β-catenin activation functions to induce cap mesenchyme-residing nephron progenitors to undergo mesenchyme-to-epithelium transformation and subsequently differentiate into nephron epithelium [145,146,147,148]. This view is supported by early gene inactivation studies, which established the requirement of Wnt4 and Wnt9b for nephron differentiation and suggested some functional redundancy with NOTCH activation [149,150,151]. Activation of β-catenin (CTNNB1) through its forced stabilization in SIX2-positive nephron progenitors induces nephron differentiation, but it has been shown to also maintain the undifferentiated nephron progenitor pool through the direct regulatory interaction with SIX2 and cooperation with MYC [136,148,152,153]. The signal activation level seems to be the key determinant of the cellular outcome of the β-catenin/WNT pathway.

Accordingly, delicate changes in signaling activity levels appear to critically dictate the fate decision in the nephron progenitor pool, as Wnt9b has also been shown to support progenitor maintenance by positively regulating proliferation [154,155]. Another UB-derived WNT ligand, Wnt11, is essential for the normal maintenance of nephron progenitors through its function in mediating the interaction between progenitors and UB tip cells [128]. The modulation of WNT signaling activity through R-spondins 1 and 3 is likely involved in signaling-level regulation, but the exact mechanisms remain to be studied as inactivation of R-Spondins only has a mild effect on progenitor propagation [156]. The dominance of the WNT/β-catenin pathway has been challenged in studies revealing the expression of NFAT and Ca^2+^ signaling components throughout nephrogenesis [157,158,159], but their exact roles await further functional evidence. 

### 7.4. FGF induced Receptor Tyrosine Kinase Signaling

The nephrogenic lineage specification and survival requires functional FGF signaling [160]. An in vitro screen of ligands that support nephron progenitors suggested that FGFs 1, 2, 9, and 20, as well as epidermal growth factor (EGF), which may require cooperative function with FGFs, can promote their proliferation [161]. Genetic inactivation studies indicate that signaling downstream of FGF receptors 1/2, specifically induced by FGF9 and -20, continues to be critical for the maintenance of self-renewal as inactivation of the ligands Fgf1 and -2 alone or in combination does not affect nephron progenitors [160,162,163,164]. Similarly, as shown for the UB lineage, the negative RTK regulator SPROUTY1 functions to balance an appropriate level of positive signaling in the nephrogenic niche to control progenitor stemness [119,165,166,167,168]. Population-intrinsic intracellular cascades evoked downstream of FGFRs in nephron progenitors include MAPK/ERK, MAPK/JNK and PI3K [87,121,169,170,171]. Nephron progenitor-specific inactivation of MAPK/ERK (Six2-TGC^tg/+^;Mek1^fl/fl^;Mek2^-/-^) results in impaired progenitor self-renewal and disorganization of the progenitor niche due to the diminished expression of PAX2, which is required for the maintenance of nephron progenitor identity and the normal function of ITGA8-mediated niche-to-extracellular matrix interaction [87,105,123,172]. Together with the additional defects in progression of nephron precursor differentiation, the nephron progenitor-specific MAPK/ERK inactivation phenotype shows close similarities to the loss of FGF8/9/20, supporting its essential function as a mediator of multiple FGF signaling functions that likely take place through PAX2 regulation [87,160,162].

### 7.5. Permanent Loss of Nephron Progenitors Terminates Renal Development

It has been demonstrated that synergistic actions of not only PI3K with WNT/ β-catenin signaling but also with BMP-induced JNK and FGF9 signaling ensure proper cell cycle progression and stemness in nephron progenitors [121,138,173]. However, the molecular causes leading to final nephron progenitor exhaustion in the end of organogenesis are only about to be revealed. 

Experimental nephron ablation by cryoinjury during the early postnatal period, when nephron progenitors are still present in the mouse kidney, indicates that additional nephron progenitors cannot be recruited to the injury site, and supports the view that the final nephron number is predetermined at birth in mouse kidneys [174]. Recent publications show that at least BMP-induced SMAD signaling and mammalian target of rapamycin (mTOR) activities may be involved in defining the timing of nephron progenitor loss, while proper micro-RNA composition is clearly required for their final depletion [138,175,176,177]. 

In 2015, the Oxburgh group reported that chemical inhibition of BMP/SMAD1/5 signaling by LDN-193189 results in hyperplastic kidneys with more nephrons than in vehicle-treated kidneys [138]. Despite their identification of a synthetic nephron progenitor niche capable of progenitor propagation, BMP inhibition is not in use for long-term nephron progenitor cultures or in differentiation protocols of stem cell-derived kidney organoids [138,178,179,180]. 

Genetic reduction of the mTOR inhibitor Hamartin (Tsc1) dosage has been shown to maintain NP cells one day longer than in control mice and increase nephron endowment by 25% [175]. More dramatic postnatal maintenance of nephron progenitors was detected in mice overexpressing RNA-binding protein Lin28, which regulates the expression of a variety of genes either by directly binding to mRNAs or by blocking the processing of the Let7-family of microRNAs [177]. Accordingly, the suppression of Let-7 itself both prolongs the nephron progenitor lifespan significantly and results in improved kidney functional parameters [176]. These experiments suggest that the abolished regulation of mRNA levels and stability resulting in a broad increase in gene activation can overcome the normal cessation program of nephrogenesis. Despite the great potential of modulating microRNA regulation, these models show either direct tumorigenesis or are associated with increased activation of the Igf2/H19 locus, which is the most prominent oncogene in pediatric kidney cancer known as Wilms tumor [181,182]. 

## 8. The Cause of Wilms Tumors

For a long time, the only genes known to be mutated or deregulated in Wilms tumors were WT1, IGF2, and genes linked to canonical WNT signaling (CTNNB1, WTX/AMER1), but recent large-scale sequencing projects have greatly extended the number of genes linked to Wilms tumorigenesis. As an excellent recent review about the genetics of Wilms tumors is available [183], here we focus on some bigger themes and their implications for understanding the biology of Wilms tumors in relation to kidney development.

The first group of Wilms tumor genes are all expressed and directly involved in the control of the nephron progenitor cells. These genes include transcription factors WT1, CTNNB1, SIX1, SIX2, EYA1, and MYCN. A logical conclusion from this would be that disruption of normal NPC biology is a root cause of Wilms tumors. 

The second group of Wilms tumor genes are miRNA processor genes (miRNAPGs) responsible for the biosynthesis of miRNAs. This group consists of DROSHA, DICER, DGCR8, XPO5, TARBP2, LIN28, and DIS3L2. The biological rationale for their mutations in Wilms tumor is unclear. It is striking, however, that mutations in such an apparently generally important biological process cause such an explicit and tissue-specific developmental problem. It would be interesting to see if the miRNAPG mutations in Wilms tumors result in changes in specific miRNAs or miRNA families and their targets. If so, this could point to specific roles for these miRNAs in normal kidney development. 

The third theme appearing in the group of Wilms tumor genes involves the responses to DNA damage in general, or the repair of double-stranded DNA (dsDNA) damage in particular, as exemplified by CHEK2, TP53, FANCD1 (BRCA2) and FANCN (PALB2) mutations. The mutations in the genes of dsDNA damage pathways are mainly known to be involved in breast and ovarian cancer. As with the miRNAPG mutations, it is remarkable to find multiple genes in this pathway mutated in such a specific developmental disease as Wilms tumors, which could suggest that the early developing kidney, perhaps specifically the NPCs, has an unexpected sensitivity for disturbance of this process.

Not only the identification of genes mutated in Wilms tumors and their cell type or stage-specific expression patterns can be informative for understanding the biology of Wilms tumors, but the mutations found in specific genes can also hold important clues. In some cases, these genotype–phenotype correlations resemble known hot-spot mutations from other cancer types or reflect the important, known control of amino acids, like the mutations found in TP53 [184]. In other cases, the rationale behind the specific mutations found in Wilms tumors remains unclear. For instance, all mutations found in SIX1 and SIX2 are Q177R mutations. This, on its own, already suggests that these are not simple loss-of-function mutations, as most loss-of-function mutations in SIX1 cause branchiootic syndrome (BOS) type 3, which is characterized by second branchial arch anomalies and ear malformations causing hearing loss but no Wilms tumors or other renal abnormalities [185]. SIX1 and SIX2 encode transcriptional regulators, and further analysis of the SIX1-Q177R mutation suggested that it results in a relaxation of its sequence-specific DNA binding and expression of additional target genes not activated by wild type SIX1 [184]. Likewise, the mutations in CTNNB1 have a surprising preference for a specific serine residue mutation [184,186]. Serine 45 is one of the four residues involved in controlling the stability of β-catenin—the protein encoded by CTNNB1—but the reason serine 45 is preferentially hit in Wilms tumor and not the other three residues that are mutated in many other cancer types is not known. Elucidating the reasons for such specific Darwinian mutational selections in Wilms tumors will not only help us understand the causes of Wilms tumors and maybe provide leads for new therapeutic opportunities but will also provide unique genetic entry points into the molecular mechanism of the proteins involved in general and in kidney development in particular.

## 9. The Origins of Wilms Tumors

Not all Wilms tumors are equal. Different histological types are linked to different genes; some mutations can be preferentially found in combination with certain other mutations, and some of these mutations can be initiating mutations while others could be involved in tumor progression [183]. Careful functional analysis of the specific mutations found in Wilms tumors in the context of a developing kidney, using animal and/or organoid models, is essential for understanding the biology of Wilms tumors and its implications for normal kidney development.

However, the genetics of Wilms tumors is only part of the story. Another essential aspect is the identification of the cell type or developmental stage in which Wilms tumor mutations occur and are selected for, as this provides the biological framework for tumorigenesis. Many lines of evidence point to disruption of the nephrogenic lineage as the primary defect leading to Wilms tumors. First and foremost, the nephrogenic rests believed to be precursor lesions of Wilms tumors histologically resemble the cell types and structures that are normally found only in developing kidneys [187]. Previous expression analyses have identified the early stages of the nephrogenic lineage as the origin of Wilms tumors [188]. More extensive expression profiling suggested that different, clinically distinct subtypes could be traced back to different developmental stages of the nephrogenic lineage [189]. Accordingly, when we mutated Wt1 in different stages of nephron development in conditional knockout mice, we found that the resulting genome-wide expression patterns resembled different clinical subtypes, with pre-MET inactivation of Wt1 resulting in expression patterns resembling WT1-mutant tumors (including signs of ectopic tissue development), while post-MET inactivation resembled WT1-wild type tumors [190]. All this, together with the identification of mutations in many genes, expressed and essential for the early nephrogenic lineage makes a very strong case that disturbance of these cells is the primary defect which initiates Wilms tumorigenesis. This does not mean, however, that every Wilms tumor has the same developmental stage of origin. It is very possible that tumors resulting from different classes of mutations, as discussed above, have different developmental origins, even within the nephrogenic lineage.

Another way to look at the origin of Wilms tumors is through its cancer stem cells. The Dekel laboratory showed that Wilms tumor cancer stem cells are defined by the expression of NCAM1 in combination with activity for the AldeFluor™ assay (STEMCELL Technologies, Cologne, Germany). Injection of only 200 double-positive cells in nude mice was sufficient for tumor formation and could recapitulate the full complexity of the original tumor [191]. The identification of Wilms tumor cancer stem cell markers is important from a therapeutic point of view, as the authors showed that treating transplanted mice with cytotoxic drugs conjugated to NCAM1 antibodies could efficiently eradicate transplanted tumors. Subsequently, it was shown that continued passaging of these xenografts enriches the blastemal component of the original tumor, which uniformly expresses SIX2 [192]. This is in accordance with an early nephrogenic lineage origin and suggests that the Wilms tumor cancer stem cell is a mutant version of the normal nephron progenitor cell found in the cap mesenchyme of the developing kidney.

At present, the same caveats regarding the potential differences between distinct Wilms tumor classes exist as discussed before. These depend on the exact initiating mutation and could also apply to the origin (or even existence) of Wilms tumor cancer stems cells, but a better understanding of the origin of these cells could be an important factor in understanding the biology of Wilms tumors. Interestingly, the possibility to make organoids from adult epithelial tissues was recently also applied to the kidney for the generation of ‘tubuloids’, while use of Wilms tumor tissue with the same protocol resulted in ‘tumoroids’ [193]. It was shown that tubuloids derived from the healthy kidney tissue of Wilms tumor patients were negative for SIX2 expression, while the tumoroids from the same patients showed high SIX2 expression. It would be interesting to see if a Wilms tumor cancer stem cell population (NCAM^+^/Aldefluor^+^ or other) is involved in the formation of these tumoroids and if this technique can be used as an in vitro alternative to identify such cells.

Lately, data have been accumulating for further nuancing the idea that the Wilms tumor cell of origin is simply a nephron progenitor cell that picked up a Wilms tumor initiating mutation(s). For instance, a careful analysis of Wilms tumor samples for markers of different embryonic kidney cell types suggested that UB cells can also be found in the tumors [194]. The same was found by Young et al. [195], who compared single-cell RNA-seq data from different renal cancer types, including Wilms tumors, to cells from normal kidneys, including embryonic kidney. This confirmed the developmental origin of Wilms tumors but also identified UB cell-specific gene expressions in the tumors. Equally intriguing is the observation that in mouse models with an oncogenic activation of β-catenin in the stromal lineage, the nephrogenic lineage is indirectly affected, resulting in a model that more closely resembled Wilms tumors than when the activation was done directly in the nephrogenic lineage [196]. Interestingly, a similar phenomenon is observed in the gastrointestinal tract, where mutations in Lkb1 result in tumorigenesis only when stromally expressed [197].

There are several possible explanations for these observations. One explanation could be that Wilms tumors, or at least some of them, originate from an even earlier developmental stage than currently believed, like the metanephric mesenchyme when some of the Wilms tumor genes with essential NPC control roles are already expressed, and potentially even before the separation of epithelial, nephrogenic, and stromal lineages to explain the inclusion of ureteric bud cells in the tumor. Alternatively, the origin and cause of Wilms tumors could derive from disturbed communication between the lineages rather than a cell-autonomous effect of the initiating mutation. Such scenarios, and others that might be equally possible, point to the difficulties in trying to deduce the origin of Wilms tumors from the end product, the final tumor. It should be remembered that Wilms tumor-initiating mutations occur during a rapid developmental process, and even if every Wilms tumor is ‘a case of disrupted development’ [198], this disruption might not result in an immediate block. If the mutant cells are not immediately blocked, we cannot simply assume that they will continue developing as they would normally do.

Careful modelling of different mutations in their normal developing context is required to fully understand where the Wilms tumors are coming from. Animal models will be essential for this, although these are technically challenging (as discussed in [7]). Alternatively, human iPSC-derived kidney organoids with genetic aberrations mimicking those found in Wilms tumors could prove useful, but it remains to be seen whether the controlled culture conditions designed for an organoid differentiation allow a cell with a Wilms tumor mutation to do the same things as it would do in a real kidney. A combination of in vivo and in vitro/organoid approaches is likely required to fully understand the developmental origins of Wilms tumors.

## 10. Conclusions

The developing kidney remains an important model to study many aspects of mammalian organ development, and most biological pathways and processes are essential for the correct development of the organ. In particular, the control of stem and progenitor cells provides an important link between basic developmental biology, regenerative medicine, and disease. Understanding the biology of Wilms tumors and their origins will not only improve the clinical options for treatment but will also provide a natural experimental system into the behavior of these stem cells during kidney development. 

## Figures and Tables

**Figure 1 genes-12-00318-f001:**
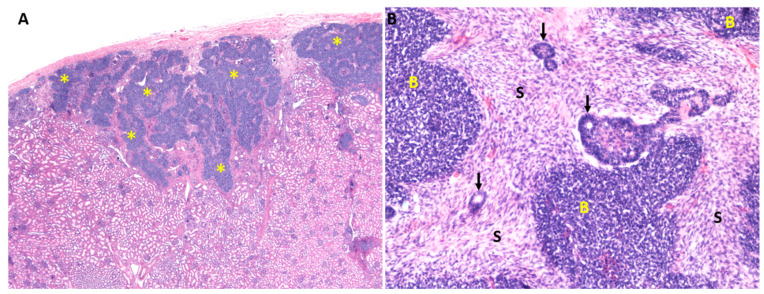
Nephroblastomatosis and Wilms tumor. (**A**) An example of human postnatal kidney with nephrogenic rests (asterisks) of the nephroblastomatosis, which are seen as tightly packed darker blue cells in the renal cortex. (**B**) An example of human kidney with the classical morphology of Wilms tumor. It resembles embryonic kidney by exhibiting blastemal (**B**), stromal (S), and epithelial cells (arrows), which, however, fail to organize into typical tissue structures.

**Figure 2 genes-12-00318-f002:**
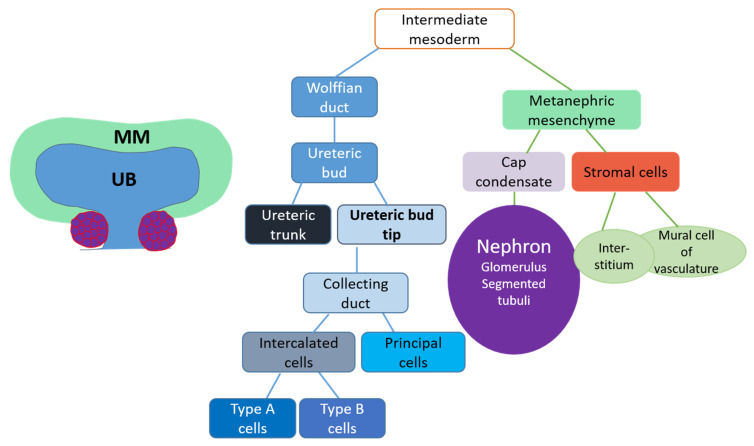
The illustration of kidney lineages and their origins. The left illustration shows a schematic presentation of an embryonic kidney with the bifurcated ureteric bud (UB, blue), which is derived from epithelial conversion of the intermediate mesoderm called the Wolffian duct. As illustrated in the middle scheme, the ureteric bud is subdivided into trunk and tip regions, where tips represent undifferentiated cells and trunks are occupied by differentiation-committed cells. Upon epithelium maturation, collecting ductal cells differentiate into specialized intercalated and principal cell types. Metanephric mesenchyme (MM, green), which surrounds the epithelial UB, lineages are depicted on the right. Metanephric mesenchyme is composed of cap condensate mesenchyme, which contains nephron progenitors, and stromal cells. The nephron progenitors in the cap condensate undergo mesenchyme-to-epithelium transition to initiate differentiation of all the segments of nephron (glomerulus and segmented tubuli) in the armpits of UB. The stromal cells of MM differentiate into renal stroma lineages.

**Figure 3 genes-12-00318-f003:**
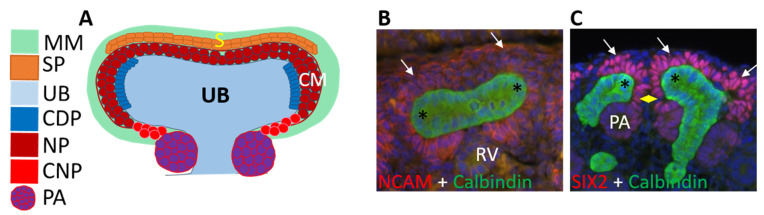
The illustration of embryonic kidney’s stem cell populations. (**A**) Schematic presentation of different kidney stem cells in their innate niches that are present during the mammalian renal organogenesis. Bifurcated ureteric bud (UB, light blue) has two tips that host collecting duct progenitors (CDP, dark blue rectangles). Immediately adjacent to the epithelial ureteric bud are nephron progenitors (NP, dark red circles), which are located in the cap mesenchyme (CM) compartment of metanephric mesenchyme (MM, green). Nephron progenitors’ stemness depends on their localization in their relationship with ureteric bud tips. Nephron progenitors that are in the armpits of the ureteric bud represent more differentiation-committed nephron progenitors (CNP, red circles), which can shuffle back and forth of the nephron progenitors in the cap mesenchyme compartment (NP, dark red circles). Fully committed nephron progenitors end up to pretubular aggregates (PA, purple balls), which are the first precursor forms of differentiating nephrons. The most cortical metanephric mesenchymal (MM, green) cells are stromal (S) cells, which also contain the stromal progenitors (SP, yellow-brownish rectangles) surrounding the outermost layer of nephron progenitors. (**B**) Embryonic kidney at day 14.5 of mouse development stained with NCAM (red), which labels all nephron progenitor and stromal progenitor cells of the metanephric mesenchyme. CALBINDIN staining (green) visualizes ureteric bud epithelium. Arrows point to an approximate nephron-to-stromal progenitor boundary, asterisks mark ureteric bud (green) tips where the collecting duct progenitors reside, and PA demarcates pretubular aggregate. (**C**) Embryonic kidney at day 14.5 of mouse development stained with SIX2 (pink), which specifically visualizes nephron progenitors only and calbindin (green) labels ureteric bud epithelium. Arrows point to the most cortical nephron progenitors, which are in direct contact with the surrounding stromal progenitors visualized by nuclear Hoechst staining (blue). Asterisks mark ureteric bud (green) tips where the collecting duct progenitors reside, diamond points to the committed nephron progenitors, PA demarcates pretubular aggregate and RV is renal vesicle.

## Data Availability

Data sharing not applicable. No new data were created or analyzed in this study. Data sharing is not applicable to this article.

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
