# Peer review of "Embryonic Kidney Development, Stem Cells and the Origin of Wilms Tumor"

_genes, 2021, doi:10.3390/genes12020318_

Round 1

Reviewer 1 Report

This review article is a detailed description of the cellular and molecular understanding of the developing kidney with a link at the end to the origin of the pediatric renal neoplasia Wilms’ tumour to cell types evident during development. While quite comprehensive and refreshingly inclusive in the material cited, there are a few aspects that I think could be improved for balance and clarity. I also feel that perhaps the review is a little heavy on kidney development, including aspects of little relevance to Wilms’ tumour, and a little light on in terms of the extensive analysis of this condition in recent years. There is also a focus throughout on the nephron progenitor population when there is growing evidence that this is not necessarily the origin of Wilms’ tumour.

Working from the abstract onwards, the use of the term embryonic stem cell to describe progenitor population in the developing kidney is inappropriate and particularly confusing when this term is really not used again but is right up front in the abstract. The term embryonic stem cell specifically applies to the pluripotent cells of the inner mass of the embryo. It should not be used to describe transient organ specific stem/progenitor cells. Indeed, while there is definitive evidence that the nephron progenitors and the stromal progenitors do self-renew, this has not been definitively proved for the ureteric epithelium. The term progenitor should be used in the abstract as for elsewhere.

The review sets a scene where there is repeatedly reference to three progenitor states in the developing kidney: nephron , stromal and ureteric progenitors. The source of the endothelium is never discussed in this review and yet there is data showing the presence of vasculogenic precursors in the organ from very early. Why omit this population from all discussion?

In the introduction, it is specifically stated that nephron progenitors are turned into Wilms’ tumour. I would contest this interpretation. It is stated later that the genetic basis, histopathological presentation and transcriptional profile varies considerably between Wilms tumours and as such it is much more likely that a developmental state somewhere between metanephric mesenchyme (hence the presence of stromal elements) and early nephron is disturbed to give Wilms’ tumour. While this is briefly discussed later on in the context of recent work ablating WT1 in stroma versus NP, stating this so definitively up front is misleading. Indeed, no citation is provided.  

While there are extensive sections discussing in great detail UB formation, branching and GDNF/RET signalling, a topic of importance for kidney development but not for Wilms’ tumour, there is a single relatively uninformative paragraph about the stroma and nothing about the origin of the blood vessels. While these are less well understood, the presence of distinct transcriptional distinctions between cortical and medullary stroma is well known and the lab of Carroll has recently investigated in some detail the stromal compartment. There has also been key lineage analyses based on Foxd1 performed by Kobayashi et al that clearly show no contribution of the stroma to nephrons and evidence for a self-renewing phenotype. This has been overlooked.

At the very beginning of the section on nephron progenitors, the term metanephric mesenchyme is used. This is almost the only time it is referred to, and yet it would be useful for the reader to understand that the metanephric mesenchyme is not just the nephron progenitors and indeed it may be a MM progenitor that forms Wilms’ tumour. A diagram of lineage relationships may help here.

The section on the molecular aetiology of Wilms’ tumours is disappointingly brief. There have now been quite a number of studies looking at transcriptional profiling that needed a more thorough discussion given the key question relates to what progenitor cell types gives rise to WT. The variation in histological features in this tumour, while mentioned briefly at the beginning, could well be expanded on here given the distinct differences between blastemal, stromal and epithelial cell types as well as the common misdifferentiation to muscle and even cartilage. Such muscle containing Wilms’ tumours have been associated with loss of WT1 function (Miyagawa er al, 1998). The presence of a triphasic histology might suggest an earlier progenitor able to give rise to more different endpoints (such as the metanephric mesenchyme). A more blastemal histology may be more indicative of an NP-derived tumour. It should also be noted that while nephrogenic rests have been regarded as preneoplastic lesions for Wilms’ tumour, they have been detected without tumours and the location of the rests has been proposed to indicate the timing of the disruption. All in all, I would like to have seen this section extended and deepened perhaps at the expense of some of the earlier development discussion.

There is a final comment about iPSC-derived organoids. It is not really clear why this is or needs to be included. I think the suggestion is that seeding a kidney organoid with WT cancer stem cells may be informative. I think we have a way to go to be sure that the environment in an iPSC-derived organoid is a reasonable model.

Minor errors:

Page 2:

‘An integral part of the safety assessment is a thorough assessment…’

‘filtrating units’ should either be filtering units or filtration units

Page 5.

Formation of the UB is not an invagination process. It is a swelling and then branching event.

‘adjecent’ should be ‘adjacent’

Page 6. …events mainly observed in cultured kidneys [64]. This should also cite 42 which clearly shows trifuractions as well as bifurcations and characterises in vivo and not in vitro tissue.

Page 7. ‘It is clear from the chimera experiments that the wild-type-derived epithelial cells are competent to populate UB tips and trunks while RET signaling-deficient cells failed to settle in the tips.’ This sentence should cite the work of Frank Costantini who performed this beautiful chimera study.

Page 7. There is an entire section describing the GDNF/RET pathway that fails to cite the work of Sanjay Jain who generated very specific mutations of key RET phosphorylation sites to dissect the role of MAPK/ERK. This section would also refer to the work of Costantini and colleagues in the GDNF-driven ablation of cam mesenchyme.

Page 10. Change ‘cycle’ to ‘cycle’

Page 10. ‘..cycle faster and are susceptible to nephron induction.’

Page 11. Change Accrodingly to Accordingly.

Page 12. The role of non-canonical Wnt signalling in nephron induction should also cite: Tanigawa S, Wang H, Yang Y, Sharma N, Tarasova N, Ajima R, Yamaguchi TP, Rodriguez LG, Perantoni AO. Wnt4 induces nephronic tubules in metanephric mesenchyme by a non-canonical mechanism. Dev Biol. 2011 Apr 1;352(1):58-69. 

Page 13: Reference relating to media that have been described to support nephron progenitors should include: Tanigawa S, Taguchi A, Sharma N, Perantoni AO, Nishinakamura R. Selective In Vitro Propagation of Nephron Progenitors Derived from Embryos and Pluripotent Stem Cells. Cell Rep. 2016 Apr 26;15(4):801-813.

Page 13. Note that most of the genes known to be mutated in Wilms’ tumours are also expressed in the metanephric mesenchyme.

Only 2 Figures are provided and their placement seems strange. Figure 2 is hard to view as both NP and CNP (not a term I like as a committed NP is a pretubular aggregate) are both red. The epithelium of the UB is also invisible with the exception of the CDP, another term that is inaccurate as the collecting duct is actually a mature state that is one of several that can arise from ureteric epithelium. In panel B, the term PA (pretubular aggregate) is sitting right on top of a structure that looks polarised and I would define as a renal vesicle (RV).

Author Response

Reviewer #1

This review article is a detailed description of the cellular and molecular understanding of the developing kidney with a link at the end to the origin of the pediatric renal neoplasia Wilms’ tumour to cell types evident during development. While quite comprehensive and refreshingly inclusive in the material cited, there are a few aspects that I think could be improved for balance and clarity. I also feel that perhaps the review is a little heavy on kidney development, including aspects of little relevance to Wilms’ tumour, and a little light on in terms of the extensive analysis of this condition in recent years. There is also a focus throughout on the nephron progenitor population when there is growing evidence that this is not necessarily the origin of Wilms’ tumour.

Response:

                      We would like to thank the reviewer for thorough, in-depth and very helpful review of our manuscript. Many points, detailed below, have made significant contribution to improve our revised version of the manuscript, which we hope the reviewer also finds suitable for publication in Genes.

Working from the abstract onwards, the use of the term embryonic stem cell to describe progenitor population in the developing kidney is inappropriate and particularly confusing when this term is really not used again but is right up front in the abstract. The term embryonic stem cell specifically applies to the pluripotent cells of the inner mass of the embryo. It should not be used to describe transient organ specific stem/progenitor cells. Indeed, while there is definitive evidence that the nephron progenitors and the stromal progenitors do self-renew, this has not been definitively proved for the ureteric epithelium. The term progenitor should be used in the abstract as for elsewhere.

Response:

                      We fully agree with the reviewer and apologize for misleading wording of embryonic stem cells in the Abstract. We have now replaced the “stem” word also elsewhere in the manuscript when it is referring to transient, embryonic kidney residing progenitors.

The review sets a scene where there is repeatedly reference to three progenitor states in the developing kidney: nephron , stromal and ureteric progenitors. The source of the endothelium is never discussed in this review and yet there is data showing the presence of vasculogenic precursors in the organ from very early. Why omit this population from all discussion?

Response:

                      We agree and acknowledge the lack of discussion on the source of the endothelium, which is stated in the “Embryonic kidney” chapter of revised manuscript (lines 89-91).

                      While we feel that the origin of endothelium is very important topic, we have to recognize that it is not our core expertise field. To better highlight the importance, we provide more references to vascular endothelium in line 91 and hope that the reviewer accepts our way of directing the readers to the articles published by the experts in this specific field.                

In the introduction, it is specifically stated that nephron progenitors are turned into Wilms’ tumour. I would contest this interpretation. It is stated later that the genetic basis, histopathological presentation and transcriptional profile varies considerably between Wilms tumours and as such it is much more likely that a developmental state somewhere between metanephric mesenchyme (hence the presence of stromal elements) and early nephron is disturbed to give Wilms’ tumour. While this is briefly discussed later on in the context of recent work ablating WT1 in stroma versus NP, stating this so definitively up front is misleading. Indeed, no citation is provided.

 Response:

                      We have revised the indicated section in the introduction and additionally included references to back up our statements. Now this runs:” Nephrons - the functional filtration units of mammalian kidneys, and renal stroma are derived from metanephric mesenchyme containing distinct progenitor pools for both lineages [5,6]. Metanephric tissue is normally lost before birth in humans but remains part of the undifferentiated nephrogenic rests in Wilms tumor patients [7,8].”

                      The reviewer is correct that in the original version of our manuscript the Introduction referred to the nephron progenitors as the cells of origin of the Wilms tumors. This has now been corrected to ‘kidney progenitors’ to indicate this might no longer be the preferred model. Further in the manuscript we extensively discuss the reasons why the nephron progenitors might or might not be the cells of origin of Wilms tumors. We agree with the reviewer that there are increasingly data appearing arguing against these (and we discuss these data), but we do not agree with the reviewer that this                means the metanephric mesenchyme must bet the origin. Some of the data would indeed be explained by a MM origin of the tumors, but other data would not. For instance, the Wilms tumor-like lesions found by stromal activation of β-catenin (Carroll lab, ref 197) would point to a post MM origin (still different from nephron progenitors though), whereas the UB components in the tumours as described by the Reeve (ref 195) and Behjati (ref 196) groups would suggest an even earlier origin than MM, namely before the separation of the nephric duct (more intermediate mesoderm-stage). This, together with other complications we discuss, like potential differences between Wilms tumor genes, different stages the mutations can occur and the complications when trying to deduce the origin of the tumor from its final histology means we don’t think it is possible at the moment to identify the MM or any other stage / cell type as the definite origin of Wilms tumors. In our opinion the section ’The                      origins of Wilms tumors’ (lines 458-537) gives a balanced and accurate description of the current ideas, possibilities and uncertainties.

While there are extensive sections discussing in great detail UB formation, branching and GDNF/RET signalling, a topic of importance for kidney development but not for Wilms’ tumour, there is a single relatively uninformative paragraph about the stroma and nothing about the origin of the blood vessels. While these are less well understood, the presence of distinct transcriptional distinctions between cortical and medullary stroma is well known and the lab of Carroll has recently investigated in some detail the stromal compartment. There has also been key lineage analyses based on Foxd1 performed by Kobayashi et al that clearly show no contribution of the stroma to nephrons and evidence for a self-renewing phenotype. This has been overlooked.

Response:

                      We have extended the section on the stromal progenitors and lineage in the revised   manuscript. Instead of focusing on the different cell types in this lineage and their function, which is not the topic of this review, we have focused on the way the stroma interacts with the other lineages and how this is starting to provide insights in the coordination of kidney development. Most likely our understanding here is barely scratching the surface, so we do not yet want to make overly definitive statements.

At the very beginning of the section on nephron progenitors, the term metanephric mesenchyme is used. This is almost the only time it is referred to, and yet it would be useful for the reader to understand that the metanephric mesenchyme is not just the nephron progenitors and indeed it may be a MM progenitor that forms Wilms’ tumour. A diagram of lineage relationships may help here.

Response:

                      This is an excellent suggestion to include diagram of different renal lineages during kidney development! We have now included this as a figure 2 in our revised manuscript.

                      We would like to point to the reviewer that we actually are discussing about metanephric mesenchyme and metanephros also in the section called “Embryonic kidney”. To give a better understanding of MM to the readers, we have modified this section as follows: “Renal stroma is a part of the mesenchymal population that caps the nephron-forming mesenchyme and is critical not only for the formation of mesangial cells and interstitium, but also actively participates in regulation of branching morphogenesis, proper differentiation of nephrons and vasculature [19-23].

The section on the molecular aetiology of Wilms’ tumours is disappointingly brief. There have now been quite a number of studies looking at transcriptional profiling that needed a more thorough discussion given the key question relates to what progenitor cell types gives rise to WT. The variation in histological features in this tumour, while mentioned briefly at the beginning, could well be expanded on here given the distinct differences between blastemal, stromal and epithelial cell types as well as the common misdifferentiation to muscle and even cartilage. Such muscle containing Wilms’ tumours have been associated with loss of WT1 function (Miyagawa er al, 1998). The presence of a triphasic histology might suggest an earlier progenitor able to give rise to more different endpoints (such as the metanephric mesenchyme). A more blastemal histology may be more indicative of an NP-derived tumour. It should also be noted that while nephrogenic rests have been regarded as preneoplastic lesions for Wilms’ tumour, they have been detected without tumours and the location of the rests has been proposed to indicate the timing of the disruption. All in all, I would like to have seen this section extended and deepened perhaps at the expense of some of the earlier development discussion.

Response:

                      We refer here to our discussion before on why in our opinion at the moment no more definite statements should be made about the origin and aetiology of the tumors. Since the Hohenstein et al 2015 review (ref 7) no new data has been published on things like the muscle differentiation in WT1-mutant tumors, and other issues the reviewer mentions, so as our current review already has 199 references we think referring to this earlier review is sufficient.

There is a final comment about iPSC-derived organoids. It is not really clear why this is or needs to be included. I think the suggestion is that seeding a kidney organoid with WT cancer stem cells may be informative. I think we have a way to go to be sure that the environment in an iPSC-derived organoid is a reasonable model.

Response:

                      We did not intend to refer to the possibility of seeding a kidney organoid with WT CSCs (although the idea is an interesting one), but to the possibility of genetically modelling Wilms tumor mutations in organoids. We agree with the reviewer that current organoid models might not be good enough for this purpose, and this is exactly what we argue in lines 523-527. Having said that, while our manuscript was under review a preprint was published on bioRxiv exactly doing this (modelling WT1-mutations in human kidney organoids; https://www.biorxiv.org/content/10.1101/2021.02.02.429313v1) and the results are better than we expected when we wrote this paragraph. As this is still preprint stage we don’t think we should already include it in our review, but of interest with respect to earlier comments by this reviewer, these organoid data are consistent with the nephron progenitor as cell type of origin.

Minor errors:

Page 2:

‘An integral part of the safety assessment is a thorough assessment…’

‘filtrating units’ should either be filtering units or filtration units

Page 5.

Formation of the UB is not an invagination process. It is a swelling and then branching event.

‘adjecent’ should be ‘adjacent’

Page 6. …events mainly observed in cultured kidneys [64]. This should also cite 42 which clearly shows trifuractions as well as bifurcations and characterises in vivo and not in vitro tissue.

Page 7. ‘It is clear from the chimera experiments that the wild-type-derived epithelial cells are competent to populate UB tips and trunks while RET signaling-deficient cells failed to settle in the tips.’ This sentence should cite the work of Frank Costantini who performed this beautiful chimera study.

Page 7. There is an entire section describing the GDNF/RET pathway that fails to cite the work of Sanjay Jain who generated very specific mutations of key RET phosphorylation sites to dissect the role of MAPK/ERK. This section would also refer to the work of Costantini and colleagues in the GDNF-driven ablation of cam mesenchyme.

Page 10. Change ‘cycle’ to ‘cycle’

Page 10. ‘..cycle faster and are susceptible to nephron induction.’

Page 11. Change Accrodingly to Accordingly.

Page 12. The role of non-canonical Wnt signalling in nephron induction should also cite: Tanigawa S, Wang H, Yang Y, Sharma N, Tarasova N, Ajima R, Yamaguchi TP, Rodriguez LG, Perantoni AO. Wnt4 induces nephronic tubules in metanephric mesenchyme by a non-canonical mechanism. Dev Biol. 2011 Apr 1;352(1):58-69. 

Page 13: Reference relating to media that have been described to support nephron progenitors should include: Tanigawa S, Taguchi A, Sharma N, Perantoni AO, Nishinakamura R. Selective In Vitro Propagation of Nephron Progenitors Derived from Embryos and Pluripotent Stem Cells. Cell Rep. 2016 Apr 26;15(4):801-813.

Response:

                      We thank the reviewer for pointing out important missing references and misspelled words in our initial submission. We have now checked the entire manuscript for English language and found some forty or so misspelled words that should be now all corrected in the revised manuscript.

                      Concerning the missing references, we have included the Short and Jain as well as both Tanigawa publications as references, while Cebrian et al paper, if the reviewer refers to this one as “GDNF-driven ablation of cam mesenchyme” on his/her comment for page 7, was similarly to chimera experiment citations from with Costantini lab already included in the initial version of our manuscript.

Page 13. Note that most of the genes known to be mutated in Wilms’ tumours are also expressed in the metanephric mesenchyme.

Response:

This is correct, and has been added to line 509-513.

Only 2 Figures are provided and their placement seems strange. Figure 2 is hard to view as both NP and CNP (not a term I like as a committed NP is a pretubular aggregate) are both red. The epithelium of the UB is also invisible with the exception of the CDP, another term that is inaccurate as the collecting duct is actually a mature state that is one of several that can arise from ureteric epithelium. In panel B, the term PA (pretubular aggregate) is sitting right on top of a structure that looks polarised and I would define as a renal vesicle (RV).

Response:

                      We have now included the third figure, which is showing the lineage commitment in developing kidney, and was suggested by the reviewer (Figure 2 of the revised manuscript).

                      We very much appreciate the reviewer’s view of terms related to kidney development. We would like to point out that committed nephron progenitors are not the same as pretubular aggregates (Lawlor et al, 2019; Elife PMID: 30676318; Lindström et al 2015; Elife PMID: 25647637) and that collecting duct progenitors are generally used definition for the ureteric bud tip cells.

                          As a response to the critiques of previous figure 2 (figure 3 of the revised manuscript), we have now changed some of the colors (orange to green), but would like to keep the others as they are to better highlight the relationship between different lineages.

Reviewer 2 Report

This is a fairly comprehensive review and I have only some minor suggestions.

  1. Page 9, line 237: I am not sure of the intended meaning of this sentence. Would it be better to delete the word "one"/
  2. There are a few spelling errors/typos:
    1. Page 11, line 314 Accordingly
    2. Page 12, line 347: exhaustion
    3. Page 15, line 435: resembled

Author Response

Reviewer #2

This is a fairly comprehensive review and I have only some minor suggestions.

  1. Page 9, line 237: I am not sure of the intended meaning of this sentence. Would it be better to delete the word "one"/
  2. There are a few spelling errors/typos:
    1. Page 11, line 314 Accordingly
    2. Page 12, line 347: exhaustion
    3. Page 15, line 435: resembled

Response:

                      We would like to thank the reviewer for the positive feedback and pinpointing the misspelled words. We have now corrected the words identified by the reviewer in the   revised manuscript. We ourselves identified additional misspelled words, which are also fixed with track changes to allow their easy observation in the revised manuscript.
